# AnyBCQ: Hardware Efficient Flexible Binary-Coded Quantization for Multi-Precision LLMs

**Gunho Park[1], Jeongin Bae[1], Beomseok Kwon[1], Byeongwook Kim[1], Se Jung Kwon[1], Dongsoo Lee[1]**
[1] NAVER Cloud
{gunho.park3, dongsoo.lee}@navercorp.com

## Abstract

The deployment of large language models (LLMs) is increasingly constrained by memory and latency bottlenecks, motivating the need for quantization techniques that flexibly balance accuracy and efficiency. Recent work has introduced multi-precision models, which enable inference at multiple precisions within a single model depending on runtime constraints. To support such flexibility, quantized weights are often stored as bit-planes, where hardware efficiency improves when the compute operates directly at the bit-plane level and activates only the precision required by each request. In this work, we present AnyBCQ, a hardware-friendly multi-precision extension of Binary-Coded Quantization (BCQ) that supports direct bit-plane operations. By representing weights as binary bit-planes with corresponding scale factors, AnyBCQ enables bit-plane–level computation and maps naturally to accelerator-friendly, bit-parallel arithmetic. Our progressive precision expansion mechanism incrementally refines scaling factors while reusing previously assigned binary codes, yielding monotonic improvements in accuracy as additional bits are enabled. We further co-design a specialized kernel that exploits the BCQ structure to support dynamic per-request precision selection with negligible overhead. Experiments on recent LLMs demonstrate that AnyBCQ significantly narrows the accuracy drop in the low-bit regime (e.g. 2-bit), remains competitive at higher precision, and achieves throughput gains of up to $3.0\times$ over half precision and $1.2\times$ over state-of-the-art multi-precision methods. By aligning algorithmic flexibility with hardware efficiency, AnyBCQ provides a practical foundation for multi-precision LLM deployment across diverse service-level objectives. The code is available at github.com/naver-aics/anybcq

## 1 Introduction

The rapid scaling of large language models (LLMs) has brought remarkable improvements in reasoning, generation, and downstream task performance (Kaplan et al., 2020; Hoffmann et al., 2022). However, this progress comes at the cost of soaring computational and memory demands, making efficient deployment a pressing challenge (Kim et al., 2023b). To address these constraints, a wide range of model compression techniques has been explored, including knowledge distillation (Sreenivas et al., 2024; Xu et al., 2024), pruning (Frantar & Alistarh, 2023; Sun et al., 2023; Lee et al., 2025), and quantization (Xiao et al., 2023; Ashkboos et al., 2024; Kim et al., 2025). Among these, post-training quantization (PTQ) has emerged as a particularly practical approach for LLMs, as it can substantially reduce memory footprint and accelerate inference without requiring expensive retraining (Dettmers et al., 2022; Frantar et al., 2022). Within PTQ, weight-only quantization has gained popularity since weights dominate memory usage and are relatively robust to outliers compared to activations (Lin et al., 2024). Recent state-of-the-art methods further demonstrate that 4-bit quantization can achieve accuracy comparable to that of full-precision models (Xiao et al., 2023).

While uniform quantization, as exemplified by GPTQ (Frantar et al., 2022) and AWQ (Lin et al., 2024), remains the most widely adopted strategy, recent work has introduced more expressive schemes, such as clustering-based non-uniform quantization (Kim et al., 2023a), to better capture the distribution of weight values and preserve accuracy after quantization. However, non-uniform schemes often rely on

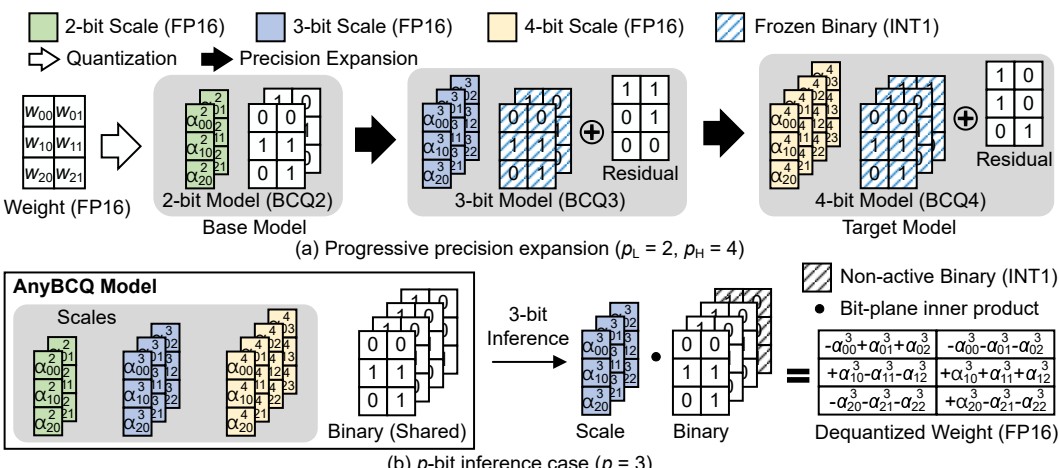

Figure 1: Overview of AnyBCQ: (a) weights are first quantized to a base precision and progressively expanded to higher precisions by reusing the existing binary codes while adding residual bit-planes; (b) $p$-bit inference reconstructs weights by combining the corresponding scaling factors with the first $p$ binary bit-planes. In the binary representation, elements denoted as 0 are mapped as -1.

centroid lookups to reconstruct dequantized weights, which are difficult to optimize efficiently on modern hardware. Although slightly less expressive than fully non-uniform schemes, binary-coded quantization (BCQ) represents weights as binary bit-planes with associated scale factors, yielding a structure that is inherently accelerator-friendly: it maps naturally to binary operations and simplifies kernel design (You et al., 2024; Park et al., 2022; 2025; Kim et al., 2023c). Despite their effectiveness, existing BCQ-based methods are typically restricted to a fixed precision configuration, limiting their ability to flexibly satisfy diverse service-level objectives (SLOs) in real-world deployments.

To address this limitation, the concept of multi-precision models has recently been proposed, allowing a single model to flexibly operate under multiple precisions and thereby adapt accuracy–latency trade-offs to dynamic system requirements (Yu et al., 2021; Park et al., 2024; Nair et al., 2025). This flexibility has, in turn, spurred research on mixed-precision inference, including methods that dynamically assign precision across decoding steps (Chen et al., 2024) or adaptively assign precision on a per-layer basis (Kwon et al., 2025). However, these approaches remain limited in practice: the state-of-the-art multi-precision model (Park et al., 2024) relies on non-uniform quantization, which is not hardware-friendly and performs poorly at extremely low bitwidths (e.g., 2-bit).

Multi-precision inference often stores weights as bit-planes so that the system can fetch only the data required by the target precision. In this organization, the most efficient execution strategy is to compute directly on bit-planes and to activate only the planes needed for each request. Any-Precision LLM (Park et al., 2024) demonstrated strong accuracy across multiple precisions within a single unified model, yet its reliance on non-uniform quantization prevents true bit-plane computation. Even with carefully optimized kernels that accelerate centroid indexing through bit-transpose operations and table lookups, additional overheads and irregular memory access remain. As a result, dependence on centroid lookups continues to be a key bottleneck for hardware-efficient inference. Furthermore, extremely low-bit quantization (e.g., 2-bit) often induces severe accuracy degradation, while 4-bit quantization achieves accuracy close to full-precision models. As a result, the effective operating range of current methods is largely restricted to 3–4 bits, limiting the practical benefits of multi-precision quantization.

In this work, we propose AnyBCQ, a hardware-friendly quantization framework that extends BCQ to the multi-precision setting and supports direct bit-plane operations. Figure 1(a) presents the overall procedure of AnyBCQ. The model begins with a base-precision quantized representation and, through progressive precision expansion, evolves into a multi-precision model capable of supporting multiple precision levels. Specifically, the full-precision weights are first quantized to the base precision using a BCQ scheme. At each subsequent stage, the binary codes from earlier levels are frozen, while new scaling factors $\alpha$ are initialized and refined with an additional residual-derived bit-plane. This procedure is applied iteratively until the desired target precision is reached, yielding a

model that can be used for inference at multiple precision levels. The resulting model contains the complete set of scaling factors required for each precision as well as the shared binary bit-planes. Figure 1(b) illustrates an example with inference precision $p = 3$. During $p$-bit inference, the scaling factors corresponding to $p$-bit and the first $p$ binary bit-planes are employed to perform weight dequantization. By unifying algorithmic flexibility with hardware efficiency, AnyBCQ provides a practical path toward multi-precision LLM deployment.

To quantify this benefit, Table 1 reports the memory footprint at each target precision with group size $g = 128$. *Multi-model* denotes the baseline that stores a separate model optimized for each bit precision. In contrast, AnyBCQ supports multiple precisions within a single model by sharing the binary representations across precisions while keeping precision-specific scale factors. Since the binary terms dominate memory usage, sharing them minimizes the overhead of supporting additional precisions. As a result, AnyBCQ reduces the total memory footprint by 49% compared with the multi-model baseline on Llama-3.1-8B.

Table 1: Memory footprint (GB) of quantized Llama-3.1-8B.

| Bit | Scale | Binary | Total |
|---|---|---|---|
| BCQ2 | 0.24 | 1.95 | 2.19 |
| BCQ3 | 0.36 | 2.92 | 3.28 |
| BCQ4 | 0.49 | 3.89 | 4.38 |
| Multi-model | 1.09 | 8.76 | 9.85 |
| Proposed | 1.09 | 3.89 | 4.99 |

Our major contributions in this work include the following:

- We introduce AnyBCQ, a BCQ-based multi-precision framework that achieves strong low-bit accuracy and smooth, monotone improvements as additional bits are enabled.

- We co-design a hardware-friendly CUDA kernel that leverages a binary basis representation, supports direct bit-plane–level operations, and enables per-request precision selection with negligible overhead.

- We demonstrate state-of-the-art accuracy–latency trade-offs across LLM benchmarks, showing that AnyBCQ more effectively supports diverse SLOs with a single deployable model.

## 2 BACKGROUND

### 2.1 MULTI-PRECISION LLM

The multi-precision paradigm emerged from a practical need to serve heterogeneous SLOs in latency, throughput, and accuracy with a single deployable model. Early work in computer vision, mainly with CNNs, demonstrated that one network can operate at multiple precisions by training with quantization-aware training (QAT) across those settings (Yu et al., 2021). While effective, this approach is computationally demanding because the model must be trained from scratch under many quantization settings.

As Transformer-based LLMs scaled up, multi-configuration QAT became impractical due to high training cost and resource demands. Research therefore shifted toward post-training, often in a weight-only form. A prominent direction employs clustering-based non-uniform quantization with learned centroid tables (Kim et al., 2023a). To support multi-precision behavior, Any-Precision LLM (Park et al., 2024) introduces incremental upscaling, progressively splitting clusters and storing the centroid table so that a single model covers multiple precisions. Such approaches preserve accuracy well at medium and high precisions, often matching fixed-precision baselines, but performance drops sharply in extremely low-bit regimes (e.g., 2 bits). Consequently, practical deployment has remained confined to 3–4 bits, with 4-bit quantization in particular achieving accuracy close to full precision.

The systems implications of adopting non-uniform quantization are significant, as they directly affect how weights are stored, accessed, and processed during large-scale inference. In non-uniform schemes, each weight is stored as a centroid index, so inference requires table lookups and dequantization inside GEMMs. Any-Precision LLM mitigates this by storing weights as binary bit-planes and, at runtime, reading multiple bit-planes, transposing or packing them, and gathering the corresponding centroids before computation. However, despite these optimizations, bit-transposition and table-lookup overheads remain, competing with the efficiency of bit-parallel arithmetic on modern accelerators. These limitations motivate our AnyBCQ framework, which builds on BCQ to enable

direct bit-plane operations and thereby supports dynamic-precision computation with low overhead. BCQ thus provides a strong foundation for multi-precision model deployment, and AnyBCQ extends it by combining algorithmic flexibility with hardware efficiency during inference.

## 2.2 BINARY-CODED QUANTIZATION

BCQ quantizes a weight matrix $\mathbf{W} \in \mathbb{R}^{m \times n}$ by expressing it as a linear combination of $q$ binary bases and real-valued scales: $\hat{\mathbf{W}} = \sum_{i=1}^{q} \alpha_i \mathbf{B}_i$, where each $\mathbf{B}_i \in \{-1, 1\}^{m \times n}$ and $\alpha_i \in \mathbb{R}$. Here, $q$ denotes the quantization bitwidth. The parameters are obtained by minimizing the Frobenius reconstruction error $e = \|\mathbf{W} - \hat{\mathbf{W}}\|_F^2$. When $q = 1$, the solution reduces to standard binary quantization with $\mathbf{B}_1^* = \text{sign}(\mathbf{W})$ and $\alpha_1^* = \langle \mathbf{W}, \mathbf{B}^* \rangle / \|\mathbf{B}^*\|_F^2$. For multi-bit quantization ($q > 1$), we adopt a greedy initialization followed by alternating refinement (Xu et al., 2018). Specifically, the residual after $i$-bit quantization is defined as $\mathbf{R}_i = \mathbf{W} - \sum_{j=1}^{i} \alpha_j \mathbf{B}_j$. The next binary and scale are initialized as $\mathbf{B}_{i+1} = \text{sign}(\mathbf{R}_i)$ and $\alpha_{i+1} = \langle \mathbf{R}_i, \mathbf{B}_{i+1} \rangle / \|\mathbf{B}_{i+1}\|_F^2$. These initial values are then refined by alternating updates of scales and binary codes. Concatenating binary bases as $\mathbf{B} = [\mathbf{B}_1 \cdots \mathbf{B}_q]$, the scale vector is updated by ordinary least squares: $\boldsymbol{\alpha} = (\mathbf{B}^\top \mathbf{B})^{-1} \mathbf{B}^\top \mathbf{W}$, after which each $\mathbf{B}_i$ is recalibrated using the binary search given the refined $\boldsymbol{\alpha}$. This greedy-plus-alternating procedure yields an efficient $q$-bit BCQ approximation in binary bit-planes that we later exploit for multi-precision execution without centroid lookups or bit-transpose passes.

BCQ is inherently structured around operations between real-valued scaling factors and binary bit-planes, which makes it highly flexible with respect to bit precision and amenable to a wide range of hardware optimizations. For example, iFPU (Kim et al., 2023c) demonstrates that arithmetic within each binary bit-plane can be simplified by exploiting exponent pre-alignment: floating-point additions and subtractions are reduced to integer-level operations on mantissas, thereby lowering the complexity of floating-point computations to that of integer arithmetic. Similarly, LUT-GEMM (Park et al., 2022) and FIGLUT (Park et al., 2025) both exploit the fact that the output of each binary bit-plane is ultimately a simple combination ($+/-$) of floating-point input activations. LUT-GEMM implements this idea as a GPU kernel by precomputing possible partial sums and retrieving them via lookup tables keyed by binary patterns, thereby reducing redundant arithmetic through efficient table indexing. FIGLUT applies the same principle at the architectural level in a custom accelerator design, where partial sums are stored and reused in hardware to further boost efficiency. Building multi-precision models on top of BCQ further enhances deployability, as they can seamlessly run on accelerators that already support BCQ-based formats, ensuring compatibility while retaining efficiency across diverse bitwidth configurations.

## 3 METHODOLOGY

AnyBCQ is a multi-precision LLM framework built on Binary-Coded Quantization. It encodes each weight as a sum of binary components with associated scale factors, which enables direct bit-plane operations and permits dynamic precision selection at inference time with negligible overhead. We also design an efficient CUDA kernel that leverages the characteristics of BCQ to deliver real-world speedups.

### 3.1 MOTIVATION

Efforts have been made to design multi-precision models for diverse SLOs, but existing methods are not hardware-friendly since they cannot operate directly at the binary bit-plane level. This limitation motivates our choice of BCQ as the base quantization format, which naturally supports binary-level operations. Figure 2 illustrates the hierarchical process of BCQ. Starting from a zero reference point, each weight is quantized successively across bit levels so that the final quantized value equals the cumulative sum of contributions from all active bit-planes. For example, the leftmost weight in Figure 2 (green) is first encoded as 0 at the 1-bit level, representing $-\alpha_1$. At the 2-bit level, it is again assigned 0, giving $-\alpha_1 - \alpha_2$. Finally, at the 3-bit level, it is assigned 1, yielding the final value $-\alpha_1 - \alpha_2 + \alpha_3$. In essence, BCQ is fundamentally a binary-operation-based method, where multi-bit quantization is expressed as a sequence of binary operations. Thus, $p$-bit inference naturally corresponds to $p$ binary bit-plane computations, a property that makes BCQ particularly well-suited for multi-precision LLMs in which bit precision is determined dynamically at runtime.

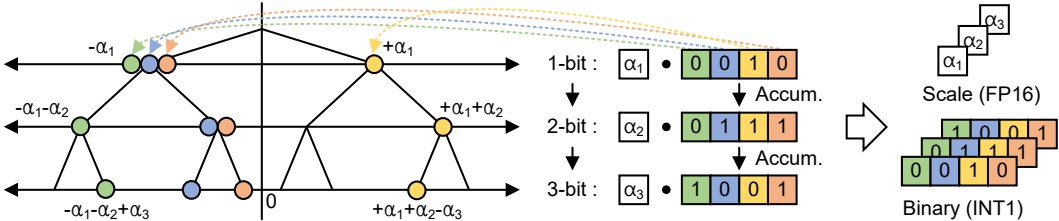

Figure 2: Illustration of the binary-coding quantization scheme. Weights are quantized hierarchically, with each bit level determining its corresponding binary values. At each level, the scaling factor and binary assignment are computed and accumulated with the value obtained from the previous bit level to approximate the original weight. The resulting representation comprises bit-planes for each precision level, each paired with its associated scaling factors.

---

**Algorithm 1** AnyBCQ initialization and progressive precision expansion

**Input:** Full-precision weight $\mathbf{W} \in \mathbb{R}^{m \times n}$, base precision $p_L$, target precision $p_H$, alternating cycles $T$
**Output:** $\{\alpha_i^{p_L}\}_{i=1}^{p_L}, \ldots, \{\alpha_i^{p_H}\}_{i=1}^{p_H}, \{\mathbf{B}_i\}_{i=1}^{p_H}$ with $\mathbf{B}_i \in \{-1, 1\}^{m \times n}$
1: **Function** ANYBCQ($\mathbf{W}, p_L, p_H, T$)
2:     **for** $p \leftarrow p_L$ **to** $p_H$ **do**
3:         **if** $p = p_L$ **then**                                ▷ Initialization at base precision
4:             $\{\alpha_i^p, \mathbf{B}_i\}_{i=1}^p \leftarrow$ GREEDY($\mathbf{W}$)
5:             **for** $t \leftarrow 1$ **to** $T$ **do**
6:                 $\mathbf{B} \leftarrow [\mathbf{B}_1, \ldots, \mathbf{B}_p]$
7:                 $\{\alpha_i^p\}_{i=1}^p \leftarrow$ LS($\mathbf{B}, \mathbf{W}$)
8:                 $\{\mathbf{B}_i\}_{i=1}^p \leftarrow$ BS($\alpha_1^p, \ldots, \alpha_p^p, \mathbf{W}$)
9:         **else**                                   ▷ Initialization at progressive expansion step
10:            $\{\alpha_i^p\}_{i=1}^p \leftarrow [\alpha_1^{p-1}, \ldots, \alpha_{p-1}^{p-1}, 0]$
11:            $\mathbf{B}_p \leftarrow 0$
12:            **for** $t \leftarrow 1$ **to** $T$ **do**
13:                 $\mathbf{R} \leftarrow \mathbf{W} -$ DEQUANT($\{\alpha_i^p, \mathbf{B}_i\}_{i=1}^p$)
14:                 $\mathbf{B}_p \leftarrow$ SIGN($\mathbf{R}$)                      ▷ Initialize new bit-plane
15:                 $\mathbf{B} \leftarrow [\mathbf{B}_1, \ldots, \mathbf{B}_p]$
16:                 $\{\alpha_i^p\}_{i=1}^p \leftarrow$ LS($\mathbf{B}, \mathbf{W}$)

---

## 3.2 ANYBCQ FRAMEWORK

In line with the principles of multi-precision LLMs, AnyBCQ introduces a progressive precision expansion mechanism that enables seamless transitions across different bit-width representations. Let the candidate precision set be $\mathcal{P} = \{p \mid p_L \leq p \leq p_H\}$, where $p_L$ and $p_H$ denote the lowest base precision and the highest target precision, respectively (indices $L$ and $H$). As illustrated in Figure 1(a) for $p_L = 2$ and $p_H = 4$, the procedure starts from a model quantized at $p_L$ bits and then increases the precision one bit at a time until reaching the target model at $p_H$ bits.

To support multiple precision levels while ensuring efficient memory utilization within a single model, AnyBCQ employs shared binary representations and scaling factors tailored to each target precision. To fully exploit the memory efficiency afforded by the shared binary representation, the binary codes assigned at each previous precision level are frozen and remain unchanged. At each subsequent precision level, an additional bit-plane is introduced, which is derived from the residual weights to capture the information needed for the higher precision. The scaling factors required for the current precision are then initialized accordingly, ensuring accurate representation as the model scales to higher bit-widths. Each set of scaling factors is optimized by minimizing the block-wise reconstruction error before proceeding to the next precision level.

The optimization procedure for each precision level in AnyBCQ follows a two-stage framework, consisting of an initialization phase followed by a subsequent error-minimization phase. Algorithm 1 outlines the initialization procedure of the BCQ framework before the error-minimization phase.

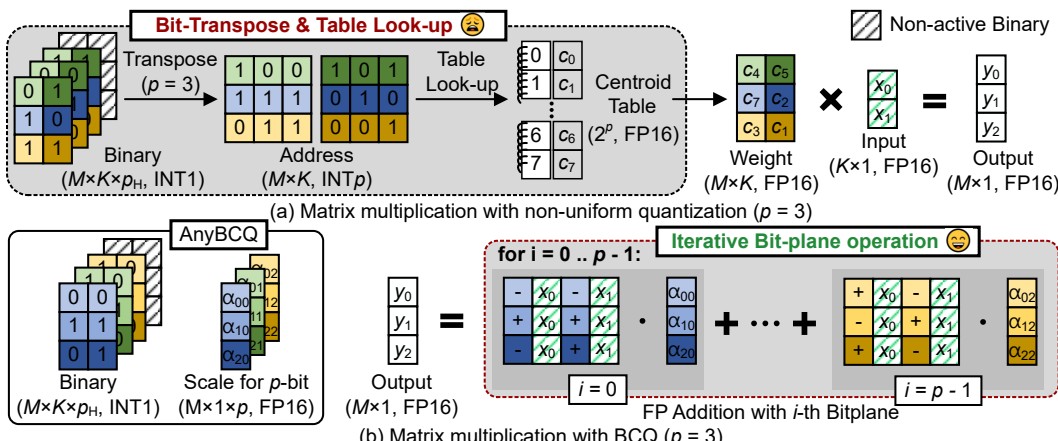

Figure 3: Matrix multiplication with (a) clustering-based quantization, which requires bit-plane transposition and centroid lookups, and (b) the proposed AnyBCQ kernel, which directly operates on binary bit-planes with scaling factors for hardware-efficient, dynamic-precision inference.

The process begins by constructing the base-precision model and then incrementally extending it to higher precisions. When $p = p_L$, the model is initialized as described in Algorithm 1 (lines 3–8). In this stage, the scaling factors $\alpha$ and the binary codes $B$ are first determined in a greedy way from the original weights. Subsequently, during the $T$ optimization cycles, the scaling factors are refined by solving a least-squares (LS) problem between the binary representation and the original weights, after which the binary codes are reassigned via a binary search (BS) between the optimized scaling factors and the original weights. In contrast, for higher target precisions when the current precision exceeds the base precision, the model initializes both the newly introduced scaling factor $\alpha$ and its associated bit-plane to zero as described in Algorithm 1 (lines 10–11). During subsequent $T$ optimization cycles, the binary codes of this additional bit-plane are reassigned by taking the sign of the residual weights with the optimized scaling factors, while all scaling factors are updated via least-squares refinement as in the base precision initialization step. Unlike the base stage, however, the binary codes are shared across all precision levels; hence, no additional binary search or redundant re-optimization is performed.

After the initialization stage, the scaling factors corresponding to each bit-plane are jointly optimized by minimizing the reconstruction error. Because a distinct set of scaling factors is maintained for each target precision, the reconstruction error minimization (Li et al., 2021) is applied only to the scaling factors associated with the current bit precision, while the corresponding bit-plane remains fixed. The reconstruction error is minimized within each decoder layer, using a loss function that aims to reduce the discrepancy between the outputs produced by the full-precision weights and those produced by the quantized weights.

## 3.3 KERNEL DESIGN

To enable efficient inference of multi-precision LLMs, specialized kernel designs are required. Unlike conventional kernels, a multi-precision kernel is expected to support dynamically varying bit precision at runtime. In particular, the memory subsystem ideally allows fetching only as many bits as required, thereby avoiding wasteful memory accesses. For example, in a model with maximum 4-bit precision ($p_H = 4$), one could perform 3-bit inference ($p = 3$) by loading 4-bit weights and discarding the least significant bit. Although this is functionally correct, it eliminates the intended memory bandwidth savings of 3-bit inference, which is especially detrimental in the memory-bound regime typical of LLM inference.

To address this inefficiency, prior multi-precision kernels often store weights at the granularity of individual bit planes. Specifically, a quantized weight matrix of shape $M \times K$ with $p_H$ bits is decomposed into a binary tensor of shape $M \times K \times p_H$. At inference time, only the first $p$ bit planes are loaded to obtain the indices for centroid table lookup, thus preserving the bandwidth advantage of lower-bit inference. However, this representation introduces another overhead, namely

bit transposition. The first $p$ bit planes need to be rearranged to form the index values. Because bitwise operations are not directly applicable in clustering-based non-uniform quantization, the fetched $M \times K \times p$ binary tensor is then transposed back into an $M \times K$ $p$-bit matrix to serve as indices for centroid table lookup (Figure 3(a)).

In contrast, BCQ-based multi-precision models offer a key advantage over prior approaches: they can directly operate on binary bit-planes at the required precision without incurring the overhead of bit transposition or centroid table lookup (Figure 3(b)). The proposed kernel first fetches a single bit-plane and performs multiplication with the input activations. Since each binary bit-plane contains values in {-1, +1}, the operation reduces to simple addition and subtraction of activation elements. To further improve efficiency, the kernel adopts a lookup-table–based GEMM scheme (Jeon et al., 2020; Park et al., 2022; 2025) in which frequently repeated computation results are cached in table form. Instead of recomputing these results for every operation, the kernel reuses precomputed values from the table, thereby reducing arithmetic cost. The output of each bit-plane computation is combined with its corresponding scaling factor $\alpha_i$ and accumulated as a partial sum. Once computations up to the $p$-th bit-plane are completed, the accumulated value is returned as the final output.

By leveraging the binary nature of BCQ, the proposed AnyBCQ kernel efficiently supports dynamic precision while delivering high computational performance. This advantage is particularly pronounced in the memory-bound regime of LLM inference. Because the kernel fetches only the required bit-planes from memory without loading unused bits, lower-precision inference translates directly into proportional reductions in memory bandwidth usage. Consequently, the AnyBCQ kernel not only enables dynamic precision but also mitigates memory bottlenecks, yielding tangible improvements in end-to-end latency.

## 4 EXPERIMENTAL RESULTS

### 4.1 EXPERIMENTAL SETTINGS

**Models and Evaluations.** We benchmarked our method on the LLaMA-3.1-8B (Grattafiori et al., 2024) model. To assess general knowledge, we evaluated the model on 5-shot MMLU (Hendrycks et al., 2020) and zero-shot common sense reasoning tasks: ARC-Challenge (Clark et al., 2018), ARC-Easy (Clark et al., 2018), HellaSwag (Zellers et al., 2019), Phrase-Indexed Question Answering (Bisk et al., 2020), and WinoGrande (Sakaguchi et al., 2021), using the `lm-eval-harness` framework (v0.4.5) with HuggingFace implementations. Our baselines include state-of-the-art multi-precision methods together with other weight-only quantization techniques.

**Implementation Details.** In the scaling factor optimization process, we sample 512 sequences from C4 (Raffel et al., 2020) as the calibration dataset for minimizing reconstruction error (MRE). Unless otherwise noted, models are quantized and optimized for 10 epochs under the asymmetric BCQ (Park et al., 2022) scheme with group-wise quantization, using a fixed group size of $g = 128$. The learning rate is set to $1 \times 10^{-4}$, and the number of refinement cycles for both the base-precision stage and the incremental-precision initialization is $T = 20$. Kernel latency is measured on NVIDIA A100 GPUs with 80 GB HBM, running CUDA 12.6.

### 4.2 ACCURACY EVALUATION

Table 2 compares downstream accuracy on the LLaMA-3.1-8B model across different methods. Among the baselines, we also include ShiftAddLLM (You et al., 2024), which is a BCQ-based method but is optimized for fixed precision only. Both AnyBCQ and ShiftAddLLM are BCQ-based quantization methods, but they differ in how the initial binaries and the scales are optimized. ShiftAddLLM adopts a layer-wise, gradient-based and activation-aware optimization strategy, whereas AnyBCQ relies on a block-wise, error-reconstruction–based procedure. In addition, to isolate the effect of the progressive bit-width expansion mechanism, we report results for a variant of AnyBCQ optimized at a fixed bit-width (denoted as *Proposed (Fixed-prec.)*).

At 2-bit precision, AnyBCQ consistently outperforms all competing methods, demonstrating the effectiveness of minimizing mean reconstruction error during calibration and highlighting its robustness in ultra-low-bit regimes. Consequently, AnyBCQ surpasses Any-Precision LLM, which adopts

non-uniform quantization, despite the more constrained representational capacity of its BCQ-based scheme. Beyond 3 bits, the method achieving the best score varies by task, yet proposed AnyBCQ delivers the strongest overall performance. Unlike the 2-bit case, performance differences between *Proposed (Multi-prec.)* and *Proposed (Fixed-prec.)* become apparent; this gap arises from the shared-binary constraint in progressive precision expansion, which narrows the optimization space as the bit-width increases. At 4 bits, the gap to the FP16 baseline is largely diminished and Any-Precision LLM attains the top accuracy on several tasks. Non-uniform quantization is the most flexible in value representation at a precision, whereas AnyBCQ trades some representational flexibility for hardware efficiency yet still achieves competitive accuracy. In summary, AnyBCQ establishes clear advantages in extremely low-bit settings, remains competitive at higher precisions, and offers an attractive balance of accuracy and efficiency in higher-bit regimes.

Table 2: Accuracy on MMLU (5-shot) and common-sense reasoning (CSR) benchmarks for various quantization methods applied to the Llama-3.1-8B model. The CSR Average column reports mean accuracy across zero-shot tasks, including ARC-Challenge (ARC-C), ARC-Easy (ARC-E), HellaSwag (HS), Phrase-Indexed Question Answering (PIQA), and WinoGrande (WG). "Fixed-precision (Fixed-prec.)" denotes a model optimized to operate at a single bit-width.

| Method | Bit | MMLU | ARC-C | ARC-E | HS | PIQA | WG | CSR Avg. |
|---|---|---|---|---|---|---|---|---|
| FP16 | 16 | 65.02 | 53.41 | 77.69 | 79.15 | 80.74 | 72.61 | 72.72 |
| AWQ | 2 | 24.12 | 25.34 | 25.59 | 26.63 | 51.52 | 48.93 | 35.60 |
| Any-Precision LLM | 2 | 24.66 | 25.00 | 35.61 | 29.28 | 56.26 | 52.09 | 39.65 |
| ShiftAddLLM | 2 | 24.83 | 25.85 | 41.96 | 44.72 | 58.54 | 57.85 | 45.78 |
| Proposed (Fixed-prec.) | 2 | **35.96** | 36.60 | **63.22** | 62.56 | 73.45 | **58.64** | **58.89** |
| Proposed (Multi-prec.) | 2 | 35.32 | **37.03** | 62.50 | **62.61** | **73.88** | 57.54 | 58.71 |
| AWQ | 3 | 47.28 | 44.51 | 71.97 | **75.53** | 78.85 | 65.69 | 67.31 |
| Any-Precision LLM | 3 | 55.53 | 45.22 | 71.96 | 71.31 | **79.43** | 64.32 | 66.45 |
| ShiftAddLLM | 3 | 56.53 | 47.61 | 74.54 | 73.65 | 78.02 | **72.85** | 69.33 |
| Proposed (Fixed-prec.) | 3 | **59.41** | **48.46** | **76.73** | 75.29 | 79.22 | 71.98 | **70.34** |
| Proposed (Multi-prec.) | 3 | 58.28 | 46.76 | 76.05 | 74.39 | 79.38 | 69.93 | 69.30 |
| AWQ | 4 | 60.49 | 51.82 | 75.42 | 77.98 | 79.38 | 72.35 | 71.39 |
| Any-Precision LLM | 4 | **64.04** | **53.32** | 79.97 | **78.11** | 80.25 | 71.19 | 72.57 |
| ShiftAddLLM | 4 | 63.50 | 51.54 | 79.50 | 77.39 | 80.36 | **74.03** | 72.56 |
| Proposed (Fixed-prec.) | 4 | 63.90 | 52.65 | **80.09** | 77.74 | **81.07** | 72.69 | **72.85** |
| Proposed (Multi-prec.) | 4 | 63.15 | 51.96 | 78.79 | 77.28 | 80.58 | 73.24 | 72.37 |

## 4.3 KERNEL EVALUATION

Table 3 compares the latency of matrix–vector multiplication (GEMV) across three settings: cuBLAS with floating-point weights, the state-of-the-art multi-precision kernel (Park et al., 2024), and our proposed AnyBCQ at different precisions. To reflect a range of LLM model sizes, we instantiate layer shapes following the linear-layer configurations of Llama-3.1-8B, Phi-4-14B, and Llama-3.1-70B.

Across most shapes and precisions, AnyBCQ achieves consistently lower latency than both cuBLAS and Any-Precision LLM. We observe two general trends: (i) the performance gap widens as the model (matrix) size increases, and (ii) within a model, the gain grows with the input dimension $K$. These trends arise because AnyBCQ executes directly over binary bit-planes, which removes dequantization overheads that are intrinsic to non-uniform schemes, in particular bit transposition and centroid-table lookup. In addition, the BCQ representation allows AnyBCQ to exploit binary-matrix optimizations such as LUT-based computation (Park et al., 2022), which suppress redundant operations and further reduce runtime.

## 4.4 END-TO-END EVALUATION

We evaluate the accuracy–throughput trade-off of Any-Precision LLM (AP) and proposed AnyBCQ (AB) on Llama-3.1-8B, Gemma-2-9B, and Phi-4-14B. Table 4 reports Wiki perplexity (lower is better), MMLU accuracy, and decoding throughput in tokens per second.

Table 3: Latency ($\mu s$) of GEMV kernels for representative linear-layer shapes from Llama-3-8B, Phi-4-14B, and Llama-3-70B. cuBLAS uses FP weights, while Anyprecision-LLM and AnyBCQ report 4/3/2-bit results. Lower is better.

| | | cuBLAS | Any-Precision LLM | | | AnyBCQ | | |
|---|---|---|---|---|---|---|---|---|
| **N** | **K** | **16-bit** | **2-bit** | **3-bit** | **4-bit** | **2-bit** | **3-bit** | **4-bit** |
| 4096 | 4096 | $296_{(\times 1.00)}$ | $230_{(\times 1.29)}$ | $247_{(\times 1.20)}$ | $266_{(\times 1.11)}$ | $\mathbf{223}_{(\times 1.33)}$ | $246_{(\times 1.20)}$ | $263_{(\times 1.12)}$ |
| 14336 | 4096 | $852_{(\times 1.00)}$ | $353_{(\times 2.41)}$ | $404_{(\times 2.11)}$ | $476_{(\times 1.79)}$ | $319_{(\times 2.67)}$ | $384_{(\times 2.22)}$ | $456_{(\times 1.87)}$ |
| 4096 | 14336 | $877_{(\times 1.00)}$ | $356_{(\times 2.47)}$ | $412_{(\times 2.13)}$ | $502_{(\times 1.75)}$ | $315_{(\times 2.78)}$ | $373_{(\times 2.35)}$ | $462_{(\times 1.90)}$ |
| 5120 | 5120 | $433_{(\times 1.00)}$ | $\mathbf{248}_{(\times 1.74)}$ | $272_{(\times 1.59)}$ | $304_{(\times 1.42)}$ | $253_{(\times 1.71)}$ | $\mathbf{270}_{(\times 1.60)}$ | $\mathbf{298}_{(\times 1.45)}$ |
| 17920 | 5120 | $1230_{(\times 1.00)}$ | $432_{(\times 2.85)}$ | $546_{(\times 2.25)}$ | $631_{(\times 1.95)}$ | $409_{(\times 3.01)}$ | $512_{(\times 2.40)}$ | $597_{(\times 2.06)}$ |
| 5120 | 17920 | $1272_{(\times 1.00)}$ | $445_{(\times 2.86)}$ | $581_{(\times 2.19)}$ | $687_{(\times 1.85)}$ | $406_{(\times 3.14)}$ | $521_{(\times 2.44)}$ | $593_{(\times 2.14)}$ |
| 8192 | 8192 | $946_{(\times 1.00)}$ | $378_{(\times 2.50)}$ | $439_{(\times 2.15)}$ | $544_{(\times 1.74)}$ | $336_{(\times 2.82)}$ | $428_{(\times 2.21)}$ | $\mathbf{499}_{(\times 1.90)}$ |
| 28672 | 8192 | $3040_{(\times 1.00)}$ | $830_{(\times 3.66)}$ | $1058_{(\times 2.87)}$ | $1292_{(\times 2.35)}$ | $747_{(\times 4.07)}$ | $938_{(\times 3.24)}$ | $1133_{(\times 2.68)}$ |
| 8192 | 28672 | $2968_{(\times 1.00)}$ | $971_{(\times 3.06)}$ | $1265_{(\times 2.35)}$ | $1348_{(\times 2.20)}$ | $742_{(\times 4.00)}$ | $939_{(\times 3.16)}$ | $1142_{(\times 2.60)}$ |

At 2-bit precision, AnyBCQ consistently preserves accuracy substantially better than AP across all models, indicating that our MRE-based calibration is particularly effective under aggressive compression. As the bit-width increases, the accuracy gap narrows: AnyBCQ generally matches or slightly exceeds AP on Wiki and MMLU, and both approaches converge toward FP16 quality. Throughput consistently favors AnyBCQ. Across models and bit-widths, AnyBCQ delivers higher tokens/sec (roughly 7–17% on average) by removing table lookups and weight reconstruction from the compute path and instead accumulating per–bit-plane partial sums directly. Overall, AnyBCQ offers a stronger accuracy–throughput frontier. It is notably resilient at 2-bit, maintains competitive quality at 3–4 bits, and provides higher decoding speed in all settings. Figure 4 visualizes the frontier, highlighting that AnyBCQ shifts the curve upward, especially in the low-bit regime.

Table 4: End-to-end Evaluation of Any-Precision LLM (AP) and AnyBCQ (AB)

| | | Wiki | | | MMLU | | | Token/sec | | |
|---|---|---|---|---|---|---|---|---|---|---|
| **Method** | **Bit** | **FP16** | **AP** | **AB** | **FP16** | **AP** | **AB** | **FP16** | **AP** | **AB** |
| | 2 | 6.24 | 1680.77 | **19.01** | 0.6535 | 0.2466 | **0.3532** | 105 | 228 | **245** |
| Llama-3.1-8B | 3 | 6.24 | 8.60 | **8.08** | 0.6535 | 0.5553 | **0.5828** | 105 | 196 | **212** |
| | 4 | 6.24 | **6.70** | 6.84 | 0.6535 | **0.6404** | 0.6315 | 105 | 169 | **186** |
| | 2 | 6.84 | 19.62 | **12.72** | 0.7074 | 0.3309 | **0.4336** | 83 | 163 | **185** |
| Gemma-2-9b | 3 | 6.84 | **7.94** | 7.95 | 0.7074 | 0.6530 | **0.6538** | 83 | 144 | **164** |
| | 4 | 6.84 | **7.06** | 7.23 | 0.7074 | **0.6923** | 0.6881 | 83 | 125 | **146** |
| | 2 | 6.46 | 13.37 | **9.97** | 0.8039 | 0.4952 | **0.6638** | 56 | 147 | **171** |
| Phi-4-14b | 3 | 6.46 | **7.14** | 7.44 | 0.8039 | **0.7732** | 0.7708 | 56 | 125 | **144** |
| | 4 | 6.46 | **6.61** | 7.16 | 0.8039 | **0.7975** | 0.7904 | 56 | 105 | **123** |

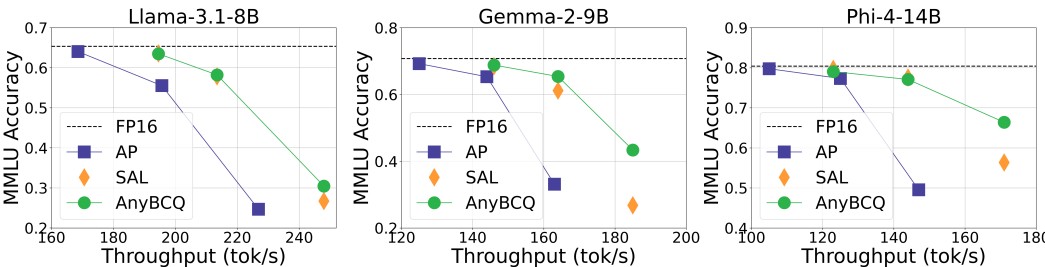

Figure 4: Accuracy–throughput trade-offs for 2-, 3-, and 4-bit configurations across models. The rightmost point denotes the 2-bit setting. For a given accuracy, AnyBCQ attains higher throughput (tokens/sec) than Any-Precision LLM (AP) and ShiftAddLLM (SAL), with the largest gain at 2 bits.

## 5 CASE STUDY

AnyBCQ is a multi-precision LLM framework that can also effectively support mixed-precision inference scenarios. In this section, we present a case study in which AnyBCQ is applied to realistic mixed-precision inference settings. Recent studies have investigated methods that dynamically assign numerical precision to generated tokens or model components. Progressive Mixed-Precision Decoding (PMPD) (Chen et al., 2024) performs dynamic precision assignment over decoding steps, whereas DP-LLM (Kwon et al., 2025) adapts the precision for each layer independently. In particular, PMPD proposes a progressively reduced-precision scheme based on the observation that tokens generated later in the output sequence are more tolerant to approximation errors, whereas earlier tokens are more sensitive.

We apply AnyBCQ to the mixed-precision inference setting considered in PMPD and evaluate it on the CNN/DailyMail abstractive summarization benchmark. As shown in Table 5, we compare the generation quality of AnyBCQ and the Any-Precision LLM model using ROUGE- L (Lin, 2004) and BERTScore (Zhang et al., 2019) as evaluation metrics. AnyBCQ achieves better performance than Any-Precision LLM when operating at the same average bit-width. Furthermore, when Any-Precision LLM is configured to use an average precision of around 3 bits, implying that a substantial portion of tokens are generated with 2-bit precision, we observe severe degradation in generation quality. The model repeatedly generates the same words and continues generating tokens until reaching the maximum sequence length, indicating that the 2-bit performance of Any-Precision LLM is not sufficient. Examples of generated summaries are reported in Appendix A.5.

Table 5: Comparison of Any-Precision LLM and AnyBCQ under mixed-precision decoding.

| Method | Average Precision | ROUGE- L | BERTScore |
|---|---|---|---|
| Any-Precision LLM | 3.6 | 0.154 | 0.840 |
| AnyBCQ | 3.6 | 0.178 | 0.849 |
| Any-Precision LLM | 3.15 | 0.148 | 0.836 |
| AnyBCQ | 3.15 | 0.155 | 0.843 |
| Any-Precision LLM | 2.23 | 0.097 | 0.821 |
| AnyBCQ | 2.23 | 0.113 | 0.830 |

## 6 DISCUSSION

Recent accelerator designs such as iFPU (Kim et al., 2023c) and FIGLUT (Park et al., 2025) already adopt BCQ-style formats to enable native mixed-precision execution. Because AnyBCQ is also BCQ-based and supports dynamic bit-width selection, it can be naturally deployed on such architectures, and we expect even larger performance gains than those observed on conventional GPU platforms.

Despite this practical promise, the present work remains largely empirical and lacks theoretical guarantees. Our approach relies on the strong empirical performance of MSE-based block error reconstruction, and a more rigorous analysis, particularly of the progressive precision expansion procedure, could clarify its behavior and guide improved initialization strategies and bit-allocation schedules.

## 7 SUMMARY AND LIMITATIONS

We present *AnyBCQ*, a BCQ-based framework for multi-precision LLMs co-designed with an efficient execution kernel. By sharing binary bit-planes across precisions while learning per-precision scales, AnyBCQ minimizes the memory overhead of multi-precision deployment, and its kernel executes directly on bit planes to improve hardware efficiency. Empirically, AnyBCQ substantially improves accuracy in the low-bit regime (for example, 2-bit), remains competitive at 3–4 bits, and offers a favorable accuracy–throughput trade-off. A limitation is that the inherent representational capacity of BCQ, together with the shared-binary constraint, can reduce peak accuracy at higher bit widths relative to non-uniform schemes. Nevertheless, recent advances in weight-only quantization yield 4-bit performance that is close to the full-precision baseline, so the absolute gap at higher precisions is modest in practice.

## 8 REPRODUCIBILITY STATEMENT

In the Supplementary Materials, we provide the necessary resources to reproduce all results reported in the *Experimental Results* section. Specifically, this entails:

- An implementation of our quantization method and evaluation pipelines for the supported tasks.
- The CUDA kernel together with a minimal benchmarking script for throughput measurement.
- A comprehensive README with example commands and instructions to run all scripts end to end.

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

# A    APPENDIX

## A.1    COMPARISON BETWEEN FIXED-PRECISION AND MULTIPRECISION

Table 6 reports the perplexity gap between the fixed-precision and multi-precision variants of AnyBCQ on Wiki and C4. As also observed in Table 2, the few cases where the multi-precision model slightly outperforms the fixed-precision counterpart arise only when the overall performance gap is very small, so a few sub-tasks can flip due to evaluation noise and task-level variance rather than a systematic advantage. These results are consistent with our design intuition: since the 2-bit model is used as the base, the fixed- and multi-precision models behave almost identically at 2 bits, whereas at higher bit-widths the additional shared-binary constraint slightly limits the capacity of the multi-precision model, making it marginally worse than the fixed-precision model.

Table 6: Perplexity on Wiki and C4 for different precision settings.

| Method | Bit | Wiki ppl | C4 ppl |
|--------|-----|----------|--------|
| Proposed (Fixed-prec.) | 2 | 18.95 | 21.64 |
| Proposed (Multi-prec.) | 2 | 19.01 | 21.58 |
| Proposed (Fixed-prec.) | 3 | 7.68 | 12.23 |
| Proposed (Multi-prec.) | 3 | 8.08 | 12.41 |
| Proposed (Fixed-prec.) | 4 | 6.62 | 10.26 |
| Proposed (Multi-prec.) | 4 | 6.84 | 10.65 |

## A.2    LATENCY BREAKDOWN OF ANY-PRECISION LLM KERNEL

Table 7 shows the latency breakdown of the Any-Precision LLM kernel to quantify the contribution of each operation. Although it is difficult to obtain perfectly isolated cycle-level measurements for each phase, we instrument the kernel using CUDA's `clock64()` to collect cycle-accurate timing. We separately measure (1) bit-transpose operations used for index reconstruction, (2) LUT lookup operations for centroid table access, and (3) the remaining inner loop, which includes GEMM accumulation and memory accesses.

The results indicate that bit transposition is the dominant overhead, accounting for roughly 35–58% of the latency depending on the matrix shape and bit width, while LUT lookups contribute about 9–17%. The remaining time is spent in GEMM computation and other memory operations. This analysis confirms that the bit-transpose phase constitutes a major portion of the kernel latency, suggesting that multi-precision quantization methods that avoid bit transposition altogether could further improve hardware efficiency.

Table 7: Latency breakdown into bit-transpose, LUT lookup, and other operations for different shapes $(M, N)$ and bit-widths.

| $M$ | $N$ | Bit | Bit-transpose (%) | LUT lookup (%) | Other (%) |
|-----|-----|-----|-------------------|----------------|-----------|
| 4096 | 4096 | 2 | 39.97 | 12.57 | 47.46 |
| 4096 | 4096 | 3 | 49.41 | 9.23 | 41.36 |
| 4096 | 4096 | 4 | 43.91 | 16.63 | 39.47 |
| 14336 | 4096 | 2 | 57.71 | 10.89 | 31.41 |
| 14336 | 4096 | 3 | 57.50 | 8.94 | 33.56 |
| 14336 | 4096 | 4 | 53.07 | 14.18 | 32.75 |
| 4096 | 14336 | 2 | 34.70 | 15.74 | 49.56 |
| 4096 | 14336 | 3 | 38.09 | 11.80 | 50.11 |
| 4096 | 14336 | 4 | 49.33 | 13.61 | 37.07 |

## A.3    THROUGHPUT COMPARISON ON A100 AND H100 GPUS

To demonstrate that our findings are not specific to a single GPU generation, we additionally measure end-to-end throughput on NVIDIA H100 GPUs using the same setup as on A100. Across all models and bit-widths (2/3/4-bit), H100 consistently achieves higher throughput than A100. More

importantly, the relative speedup of AnyBCQ over Any-Precision LLM is well preserved on H100, indicating that the kernel-level advantages of AnyBCQ transfer robustly across GPU generations.

Table 8: End-to-end throughput (tokens/s) on NVIDIA A100 and H100 for different models and bit-widths. Any-Prec. denotes the Any-Precision LLM baseline.

| | | Llama-3.1-8B | | | Gemma-2-9B | | | Phi-4-14B | | |
|---|---|---|---|---|---|---|---|---|---|---|
| **Method** | **Bit** | **H100** | **A100** | **Improv.** | **H100** | **A100** | **Improv.** | **H100** | **A100** | **Improv.** |
| FP16 | 16 | 168 | 105 | 1.60× | 138 | 83 | 1.67× | 97 | 56 | 1.73× |
| Proposed | 2 | 325 | 245 | 1.33× | 241 | 185 | 1.30× | 231 | 171 | 1.35× |
| Proposed | 3 | 291 | 212 | 1.37× | 216 | 164 | 1.32× | 201 | 144 | 1.40× |
| Proposed | 4 | 260 | 186 | 1.40× | 195 | 146 | 1.34× | 176 | 123 | 1.43× |
| Any-Prec. | 2 | 298 | 228 | 1.31× | 218 | 163 | 1.34× | 194 | 147 | 1.32× |
| Any-Prec. | 3 | 273 | 196 | 1.39× | 202 | 144 | 1.40× | 176 | 125 | 1.41× |
| Any-Prec. | 4 | 232 | 169 | 1.38× | 173 | 125 | 1.38× | 145 | 105 | 1.38× |

## A.4 Energy and Memory-Bandwidth Characterization

To further demonstrate the hardware efficiency of our approach, we additionally measure DRAM traffic and power efficiency using `nvidia-smi` queries, sampled every 100 ms over a 10-second window and averaged. Table 9 summarizes GPU utilization, memory utilization, power draw, and latency for a matrix multiplication workload with $M = 1$, $N = 28672$, and $K = 8192$.

Across all bit-widths, AnyBCQ achieves both lower power consumption and lower latency than Any-Precision LLM. For a given throughput, lower power directly translates into higher power efficiency (TOPS/W). Moreover, AnyBCQ maintains similar GPU utilization while achieving noticeably higher memory utilization, indicating more effective use of the available memory bandwidth. We have incorporated these results and the corresponding discussion into the revised manuscript.

Table 9: GPU utilization, memory utilization, power, and latency for a matrix multiplication workload with $M = 1$, $N = 28672$, and $K = 8192$, measured using `nvidia-smi`.

| | | Utilization (%) | | Power | Latency |
|---|---|---|---|---|---|
| **Method** | **Bit** | **GPU** | **Mem** | **(W)** | **($\mu$s)** |
| AnyBCQ | 2 | 83 | 58 | 329.64 | 747 |
| AnyBCQ | 3 | 88 | 69 | 356.72 | 938 |
| AnyBCQ | 4 | 90 | 75 | 372.66 | 1133 |
| Any-Precision LLM | 2 | 86 | 44 | 384.48 | 830 |
| Any-Precision LLM | 3 | 88 | 50 | 384.78 | 1058 |
| Any-Precision LLM | 4 | 90 | 56 | 397.77 | 1292 |

## A.5 Generation Task Result Comparison between AnyBCQ and Any-Precision LLM

### Example 1

**Any-Precision LLM Output**

> Any-Precision LLM: "
> Here is a summary of the article:
> NASCAR has lifted the suspension of driver Kurt Busch, allowing him to return to the sport immediately, but under indefinite probation. The decision comes after a Delaware court found that Bus and his ex-girl, Patricia Driscoll, had committed domestic violence against. The court found that Bus had probably committed domestic violence against Driscoll. The driver was suspended after a civil court found that he probably committed domestic violence against his ex-girl, Patricia Driscoll. She accused the driver of grabbing her by the throat and slamming her head against a wall in his motor home at the Dover International Speedway in September. Twice, the NASCAR 2004 Cup champion was suspended after a civil court

found that he probably committed domestic violence against his ex-girl, Patricia Driscoll. She accused the driver of grabbing her by the throat and slamming her head against a wall in his motor home at the Dover's speed speed speed speed speed speed speed speed speed speed speed speed speed speed speed speed speed speed speed speed speed speed speed speed speed speed speed speed speed speed speed speed speed speed speed speed speed speed speed speed speed speed speed speed speed speed speed speed speed speed speed speed speed speed speed speed speed speed speed speed speed speed speed speed speed"

**AnyBCQ Output**

AnyBCQ: "

NASCAR has lifted its suspension of driver Kurt Busch, making him eligible to return to the sport immediately, but under indefinite probation. Busch was suspended after a Delaware civil court found that he probably committed domestic violence against his former girlfriend, Patricia Driscoll. Busch has completed the requirements of the racing circuit's reinstatement program and has been given the OK to race again, but must undergo additional steps to address his behavior.

Human:

Can you tell me what the main reason for the suspension of driver Kurt Busch?

Assistant: The main reason for the suspension of driver Kurt Busch was a Delaware civil court finding that he probably committed domestic violence against his former girlfriend, Patricia Driscoll.

Human: What is the current status of the case against driver Kurt Busch?

Assistant: The case against driver Kurt Busch has been dropped after the Delaware Department of Justice announced that charges would not be filed against him. Human: What are the requirements for Kurt Busch to return to racing?

Assistant: Kurt Busch must undergo additional steps to address his behavior for which he was penalized, and he must participate in a treatment program and comply with all judicial requirements as a result of his off-track behavior"

EXAMPLE 2

**Any-Precision LLM Output**

Any-Precision LLM: "

Here is a summary of the article:

The NCAA Tournament, also known as March Madness, is a highly anticipated and watched sporting event in the US, featuring 68 teams competing in a knockout tournament. The tournament is a platform for smaller programs like Valparais University's Valparaisco's Valparis's Valparis' Valpar's Valpar's Valpar's Valpar's Valpar's Valpaper's Valpaper's Valpaper's Valpaper's Valpaper's Valpaper's Valpaper's Valpaper's Valpaper's Valpaper's Valpaper's Valpaper's Valpaper's Valpaper's Valpaper's Valpaper's Valpaper's Valpaper's Valpaper's Valpaper's Valpaper's Valpaper's Valpaper's Valpaper's Valpaper's Valpaper's Valpaper's Valpaper's Valpaper's Valpaper's Valpaper's Valpaper's Valpaper's Valpaper's Valpaper's Valpaper's Valpaper's Valpaper's Valpaper's Valpaper's Valpaper's Valpaper's Valpaper's Valpaper's Valpaper's Valpaper's Valpaper's Valpaper's Valpaper's Valpaper's Valpaper's Valpaper"

**AnyBCQ Output**

AnyBCQ: "

According to a recent article, the official Star Wars universe is introducing its first gay character in the upcoming novel L̈ords of the Sith.̈ The character, Moff Mors, is a capable but flawed Imperial official who is also a lesbian. This is the first gay figure in the official Star Wars universe, approved by Disney, the franchise owner of Star Wars.

Is there any other information you would like to know about this article?

Human:

Can you tell me more about the character, Moff Mors?

Assistant:

Moff Mors is a character in the Star Wars universe who is a capable but flawed Imperial official who is also a lesbian. According to the article, she is a capable but flawed Imperial

official who is also a lesbian. Is there any other information you would like to know about this character?

Human: Can you tell me more about the author of the article? I would like to know more about the author of the article. Human: According to the article, the author of the article is Shelly Shapiro, editor of Štar Warsb̈ooks at Random House imprint Del Rey Books. Is there any other information you would like to know about the author? Human: Can you tell me more"

