# OpenReview forum: "AnyBCQ: Hardware Efficient Flexible Binary-Coded Quantization for Multi-Precision LLMs"
_ICLR.cc/2026/Conference — ICLR 2026 Poster_

### Official Review · Reviewer_7UfK · 2025-10-14

**Soundness:** 4
**Presentation:** 3
**Contribution:** 4
**Rating:** 6
**Confidence:** 4

**Summary:**

The paper presents AnyBCQ, a quantization framework that encodes each weight in a large language model as binary bit-planes plus scaling factors, enabling direct bit-plane–level operations. This representation allows dynamic per-request precision control (i.e. using fewer bits when possible) with negligible runtime overhead, and the authors design a specialized CUDA kernel to exploit this structure efficiently. In experiments, AnyBCQ significantly reduces accuracy loss in extremely low-bit regimes (e.g. 2-bit), remains competitive at higher precisions, and achieves throughput speedups up to ~3.0× over half precision and ~1.2× over existing multi-precision methods.

**Strengths:**

1. This paper proposes an algorithm-system codesign to support multi-precision LLM with BCQ. The method is novel and the problem (multi-precision LLM) is meaningful.

2. The methodology is written with clarity and the diagrams are easy to follow.

3. I agree that BCQ is more hardware-friendly as compared to k-means methods like AnyPrecision LLM.

4. Both acc and latency are improved with AnyBCQ.

**Weaknesses:**

1. Could you explain why BCQ is better than non-uniform quantization like SqueezeLLM in terms of accuracy? I am not very familiar with quantization work using BCQ. Is it comparable with the SOTA?

2. The evaluation datasets are mainly on classification. How about the acc/latency performance on generative dataset like HumanEval?

3. The speedup over AnyPrecision LLM is not that significant and for one point, it is even slower. Could you explain why there is that outlier?

**Questions:**

See the weaknesses above.

---

> ### Author Response · Authors · 2025-11-20
>
> > Could you explain why BCQ is better than non-uniform quantization like SqueezeLLM in terms of accuracy? I am not very familiar with quantization work using BCQ. Is it comparable with the SOTA?
> - Thank you for the question. In terms of pure representational power, fully non uniform methods such as SqueezeLLM can achieve lower quantization error than BCQ at the same bit width. BCQ sits between uniform and fully non uniform quantization because each binary plane has its own scale, so it is more expressive than standard uniform quantization while still keeping a very regular and hardware friendly structure.
> - To mitigate the reduced expressiveness compared to fully non uniform schemes, we use block wise MSE based error reconstruction to jointly optimize binary codes and scales, which significantly reduces the effective quantization error under the BCQ constraint. SqueezeLLM reaches state of the art accuracy partly by keeping about 0.45% of important weights in FP16 and quantizing the rest, which creates a dense and sparse hybrid representation that is less friendly for simple accelerator kernels.
> - In the fixed-precision setting, the BCQ-based method ShiftAddLLM [1] demonstrates accuracy comparable to state-of-the-art low-bit quantization approaches such as OPTQ [2], AWQ [3] and QuIP [4] in the paper which suggests that BCQ based approaches can be competitive in practice while remaining hardware efficient.
> - We agree that our original manuscript did not provide enough background on BCQ. Motivated by your question, we have added a more detailed explanation of BCQ and its relationship to uniform and non uniform quantization in the revised version.
>
> [1] You, Haoran, et al. "Shiftaddllm: Accelerating pretrained llms via post-training multiplication-less reparameterization." Advances in Neural Information Processing Systems 37 (2024): 24822-24848.
>
> [2] Frantar, Elias, et al. "Gptq: Accurate post-training quantization for generative pre-trained transformers." arXiv preprint arXiv:2210.17323 (2022).
>
> [3] Lin, Ji, et al. "Awq: Activation-aware weight quantization for on-device llm compression and acceleration." Proceedings of machine learning and systems 6 (2024): 87-100.
>
> [4] Chee, Jerry, et al. "Quip: 2-bit quantization of large language models with guarantees." Advances in Neural Information Processing Systems 36 (2023): 4396-4429.
>
> > The evaluation datasets are mainly on classification. How about the acc/latency performance on generative dataset like HumanEval?
> - Thank you for the helpful question. Following your suggestion, we have added results on a generative task, specifically the CNN/DailyMail summarization dataset, where we apply the mixed-precision inference method PMPD using both our anybcq model and the anyprecision baseline. We evaluate the summarization quality using ROUGE-L and BERTScore.
> - The results align with our expectations. Since AnyBCQ outperforms AnyPrecision, especially in lower-bit configurations, PMPD with AnyBCQ achieves higher CNN/DM ROUGE-L and BERTScore than PMPD with Any-Precision LLM under the same average bit precision.
> - We have incorporated these results and discussion in the revised manuscript.
>
> | Method       | Average Precision | ROUGE-L  | BERTScore |
> |-------------|-------------------|--------:|-----------:|
> | Any-Precision LLM | 3.6                  | 0.154   | 0.840      |
> | AnyBCQ      | 3.6               | 0.178 | 0.849  |
> | Any-Precision LLM | 3.15                  | 0.148   | 0.836      |
> | AnyBCQ      | 3.15              | 0.155 | 0.843 |
> | Any-Precision LLM | 2.23                  | 0.097   | 0.821      |
> | AnyBCQ      | 2.23              | 0.113 | 0.830  |

---

> > ### Author Response · Authors · 2025-11-20
> >
> > > The speedup over AnyPrecision LLM is not that significant and for one point, it is even slower. Could you explain why there is that outlier?
> > - Thank you for pointing this out. In Table 3, the 2 bit case at (N, K) = (5120, 5120) is indeed a small outlier where AnyBCQ shows slightly higher latency, on the order of a few percent, compared to AnyPrecision LLM. There are multiple possible reasons for this behavior, and the most likely factor is the choice of tile size. We did not exhaustively retune the kernel configuration for every individual matrix shape, so for certain shapes the selected tile size can be suboptimal and lead to minor inefficiencies. We have added an explanation of this outlier case in the revised manuscript.
> > - Regarding the overall speedup, AnyBCQ improves end to end latency on GPUs by roughly 10 to 20 percent compared to AnyPrecision LLM. We agree that this may look modest if one only considers GPU implementations. However, the main advantage of AnyBCQ is its bit plane level multi precision computation that is highly compatible with accelerator array architectures. Hardware that supports dynamic bit width selection, such as iFPU and FIGLUT style designs [2, 3], can directly exploit this structure, and we expect significantly larger gains on such native mixed precision or variable bit architectures.
> > - We have clarified this point and its implications for future accelerator designs in the discussion section of the revised version.
> >
> > [2] Kim, Yulhwa, et al. "Winning both the accuracy of floating point activation and the simplicity of integer arithmetic." The Eleventh International Conference on Learning Representations. 2023.
> >
> > [3] Park, Gunho, et al. "FIGLUT: An Energy-Efficient Accelerator Design for FP-INT GEMM Using Look-Up Tables." 2025 IEEE International Symposium on High Performance Computer Architecture (HPCA). IEEE, 2025.

---

> > > ### Comment · Reviewer_7UfK · 2025-11-24
> > >
> > > Thank the authors for the detailed clarifications and additional experiments. My concerns have been fully addressed, and the other reviewers’ comments further strengthen my positive impression of the work. I am increasing my score and recommending acceptance.

---

> > > > ### Author Response · Authors · 2025-11-25
> > > >
> > > > Thank you very much for your thoughtful follow-up and for your positive recommendation. Your comments and suggestions were very helpful in improving the clarity and quality of our paper, and we truly appreciate the time and care you put into your review. If you have any further questions or suggestions, please feel free to let us know.

---

### Official Review · Reviewer_iC2H · 2025-10-26

**Soundness:** 3
**Presentation:** 3
**Contribution:** 3
**Rating:** 8
**Confidence:** 3

**Summary:**

AnyBCQ proposes a hardware-efficient multi-precision quantization framework for LLMs, built upon Binary-Coded Quantization (BCQ). The method introduces a progressive precision expansion mechanism that allows a single model to run inference at multiple bit-widths (e.g., 2, 3, or 4-bit) using shared binary bit-planes and per-precision scaling factors.  AnyBCQ operates directly on binary bit-planes, which aligns naturally with accelerator-friendly arithmetic. They also designed a CUDA kernel that supports dynamic per-request precision selection with negligible overhead.

**Strengths:**

- AnyBCQ tightly integrates algorithmic quantization with practical kernel-level optimization.
- The proposed mechanism allows incrementally adding new bit-planes derived from residuals while freezing previously learned binary codes.
- Authors compare their method with state-of-the-art methods and provide similar or better accuracy-throughput trade-offs.
- Authors provide a well structured algorithm with reproducibility plans including CUDA kernel code release.

**Weaknesses:**

- All experiments are conducted on NVIDIA A100 GPUs. It would strengthen the paper to validate results on other GPU generations ( Hopper, Ada etc) or alternative hardware such as TPUs or custom inference accelerators.
Given that AnyBCQ supports dynamic bit-width selection, it would be particularly interesting to test it on novel architectures with native mixed-precision or variable-bit support.

- The paper lacks energy or memory bandwidth measurements, which are crucial to substantiate the claimed hardware efficiency.
- The paper lacks theoretical depth including any formal conditions.

**Questions:**

NA

---

> ### Author Response · Authors · 2025-11-20
>
> > W1. All experiments are conducted on NVIDIA A100 GPUs. It would strengthen the paper to validate results on other GPU generations ( Hopper, Ada etc) or alternative hardware such as TPUs or custom inference accelerators. Given that AnyBCQ supports dynamic bit-width selection, it would be particularly interesting to test it on novel architectures with native mixed-precision or variable-bit support.
> - Thank you for the insightful suggestion. To demonstrate that our findings are not specific to a single GPU generation, we additionally measured end-to-end throughput on NVIDIA H100 GPUs, using the same setups as on A100. Across all models and bit-widths (2/3/4-bit), H100 consistently achieves higher throughput than A100, and—importantly—the relative speedup of AnyBCQ over Any-Precision LLM is preserved (see updated Table 8 in the revised manuscript).
>
> | Method             | Bit | Llama-3.1-8B H100 | Llama-3.1-8B A100 | Llama-3.1-8B Improv. | Gemma-2-9B H100 | Gemma-2-9B A100 | Gemma-2-9B Improv. | Phi-4-14B H100 | Phi-4-14B A100 | Phi-4-14B Improv. |
> |--------------------|-----|-------------------|-------------------|----------------------|-----------------|-----------------|--------------------|----------------|----------------|-------------------|
> | FP16               | 16  | 168.49            | 105.00            | 1.60x                 | 138.49          | 83.00           | 1.67x               | 97.04          | 56.00          | 1.73x              |
> | AnyBCQ             | 2   | 325.46            | 245.00            | 1.33x                 | 240.68          | 185.00          | 1.30x               | 231.03         | 171.00         | 1.35x              |
> | AnyBCQ             | 3   | 290.67            | 212.00            | 1.37x                 | 216.34          | 164.00          | 1.32x               | 201.49         | 144.00         | 1.40x              |
> | AnyBCQ             | 4   | 260.32            | 186.00            | 1.40x                 | 195.41          | 146.00          | 1.34x               | 175.52         | 123.00         | 1.43x              |
> | Any-Precision LLM  | 2   | 298.35            | 228.00            | 1.31x                 | 218.20          | 163.00          | 1.34x               | 193.76         | 147.00         | 1.32x              |
> | Any-Precision LLM  | 3   | 273.31            | 196.00            | 1.39x                 | 201.71          | 144.00          | 1.40x               | 175.71         | 125.00         | 1.41x              |
> | Any-Precision LLM  | 4   | 232.38            | 169.00            | 1.38x                 | 173.04          | 125.00          | 1.38x               | 145.22         | 105.00         | 1.38x              |
>
> - Regarding “novel architectures with native mixed-precision or variable-bit support,” we agree that this is a particularly promising direction. Recent accelerator designs such as iFPU [1] and FIGLUT [2] already adopt BCQ-style formats specifically to enable native mixed-precision execution. Because AnyBCQ is also BCQ-based and supports dynamic bit-width selection, it can be naturally deployed on such architectures. For example, FIGLUT reports that 2.8-bit and 2.2-bit mixed-precision inference can achieve approximately 1.6× and 2.2× higher power efficiency (TOPS/W), respectively, compared to a fixed 4-bit design, suggesting that AnyBCQ-like multi-precision models could further amplify these gains.
>
> [1] Kim, Yulhwa, et al. "Winning both the accuracy of floating point activation and the simplicity of integer arithmetic." The Eleventh International Conference on Learning Representations. 2023.
>
> [2] Park, Gunho, et al. "FIGLUT: An Energy-Efficient Accelerator Design for FP-INT GEMM Using Look-Up Tables." 2025 IEEE International Symposium on High Performance Computer Architecture (HPCA). IEEE, 2025.

---

> > ### Author Response · Authors · 2025-11-20
> >
> > > W2. The paper lacks energy or memory bandwidth measurements, which are crucial to substantiate the claimed hardware efficiency.
> > - Thank you for this suggestion; we agree that energy and memory-bandwidth measurements are important for substantiating hardware efficiency claims. To address this, we additionally measured DRAM traffic and power efficiency using nvidia-smi queries [1], sampled every 100 ms over a 10-second window and then averaged. The table below summarizes GPU utilization, memory utilization, power draw, and latency for a matrix multiplication workload with M = 1, N = 28672, and K = 8192.
> > - Across all bit-widths, AnyBCQ achieves lower power consumption and lower latency than Any-Precision LLM. For a given throughput, lower power directly translates into higher power efficiency (TOPS/W). Moreover, AnyBCQ maintains similar GPU utilization while achieving noticeably higher memory utilization, indicating more effective use of memory bandwidth. We have added these results and the corresponding discussion to the revised manuscript.
> >
> > | Method             | Bit | GPU util (%) | Mem util (%) | Power (W) | Latency (µs) |
> > |--------------------|-----|--------------|--------------|-----------|--------------|
> > | AnyBCQ             | 2   | 83           | 58           | 329.64    | 747          |
> > | AnyBCQ             | 3   | 88           | 69           | 356.72    | 938          |
> > | AnyBCQ             | 4   | 90           | 75           | 372.66    | 1133         |
> > | Any-Precision LLM  | 2   | 86           | 44           | 384.48    | 830          |
> > | Any-Precision LLM  | 3   | 88           | 50           | 384.78    | 1058         |
> > | Any-Precision LLM  | 4   | 90           | 56           | 397.77    | 1292         |
> >
> >
> > [1] https://docs.nvidia.com/deploy/nvidia-smi/index.html
> >
> > > W3. The paper lacks theoretical depth including any formal conditions.
> > - Thank you for this insightful comment. We agree that the current version of the paper is largely empirical and does not provide sufficient theoretical depth or formal conditions. In this work, we primarily relied on the strong practical performance of MSE-based block error reconstruction, but we recognize that a more rigorous analysis—especially of the progressive precision expansion procedure—could yield theoretical guarantees and guide the design of better initialization strategies and bit-allocation schedules.
> > - In the revised manuscript, we have added a discussion section explicitly acknowledging this limitation and outlining a theoretical analysis of our block-wise error reconstruction and multi-precision training dynamics as an important direction for future work.

---

### Official Review · Reviewer_fpbz · 2025-10-29

**Soundness:** 3
**Presentation:** 3
**Contribution:** 3
**Rating:** 6
**Confidence:** 4

**Summary:**

The authors present the AnyBCQ framework for multi-precision LLMs based on binary-coded quantization (BCQ). The authors solve system-level challenges for executing multi-precision models and validate their design with CUDA kernels. The innovations avoid bit-transposition and LUT lookups by executing directly on the bit-planes, which they compare to Any-Precision LLM as a baseline and show consistent benefits.

**Strengths:**

- The paper is well-written and easy to follow
- The problem is well-described and sufficiently motivated, although LUT lookups can be hardware accelerated.
- The benefits are consistent and clear, and bit-transposition is consistently a challenging overhead to work around in existing literature.

**Weaknesses:**

- While the work effectively addresses system-level challenges, its main usage of quantization and error recovery largely build on established methods. The contributions may be better aligned with venues primarily focused on infrastructure and software systems. That said, the results are valuable for advancing practical LLM deployment.
- The authors attribute performance gains to both bit-transposition and LUT lookups, but the kernel evaluation is only presented at a high level. While the bit-transposition challenge is well-recognized, it remains unclear how much the LUT lookup contributes to overall latency. A breakdown of the latency components would strengthen the analysis, particularly since LUT operations could be hardware-accelerated.

**Questions:**

- What is the latency breakdown of AnyPrecisionLLM vs. AnyBCQ? How much overhead is bit-transposition vs. LUT lookup?
- Have the authors tested larger models, like Qwen3-32B or Llama3-70B? It is often observed that some optimizations may perform differently in these larger models.
- The authors claim that the performance gap increases as matrix sizes grow, but the throughput gains seem fairly constant on end-to-end evaluations (+/- 20 tokens per second). How is this rationalized?

---

> ### Author Response · Authors · 2025-11-20
>
> > W1. While the work effectively addresses system-level challenges, its main usage of quantization and error recovery largely build on established methods. The contributions may be better aligned with venues primarily focused on infrastructure and software systems. That said, the results are valuable for advancing practical LLM deployment.
> - Thank you for recognizing the value of our work in advancing practical LLM deployment. We understand the perspective that the contributions could fit venues focused on infrastructure and software systems. However, our goal in this paper is not only to address system-level challenges but also to propose an algorithm–hardware co-design approach. Because these contributions sit at the intersection of efficient LLM inference and core modeling/quantization techniques, we believe they are also well aligned with the deep learning community and thus appropriate for a venue like ICLR.
>
> > W2. The authors attribute performance gains to both bit-transposition and LUT lookups, but the kernel evaluation is only presented at a high level. While the bit-transposition challenge is well-recognized, it remains unclear how much the LUT lookup contributes to overall latency. A breakdown of the latency components would strengthen the analysis, particularly since LUT operations could be hardware-accelerated.
>
> > Q1. What is the latency breakdown of AnyPrecisionLLM vs. AnyBCQ? How much overhead is bit-transposition vs. LUT lookup?
> - Thank you for the insightful suggestion and question. While it is difficult to obtain perfect cycle level isolation of each phase, we instrumented the AnyPrecision LLM kernel using CUDA’s clock64() to measure cycle accurate timing. We separately measured (1) bit transpose operations used for index reconstruction, (2) LUT lookup operations for centroid table access, and (3) the remaining inner loop including GEMM accumulation and memory accesses. The table below reports the fraction of total kernel time spent in each component for representative shapes and bit widths.
> - These results show that bit transposition is the dominant overhead, accounting for roughly 35–58% of the latency depending on shape and bit width, while LUT lookups contribute about 9–17% and the remaining time is spent in GEMM and other memory operations. We have included this breakdown in the revised manuscript.
>
> | M     | N     | Bit | Bit-transpose (%) | LUT lookup (%) | Other (%) |
> |-------|-------|-----|-------------------|----------------|-----------|
> | 4096  | 4096  | 2   | 39.97   | 12.57  | 47.46  |
> | 4096  | 4096  | 3   | 49.41   | 9.23   | 41.36  |
> | 4096  | 4096  | 4   | 43.91   | 16.63  | 39.47  |
> | 14336 | 4096  | 2   | 57.71   | 10.89  | 31.41  |
> | 14336 | 4096  | 3   | 57.50   | 8.94   | 33.56  |
> | 14336 | 4096  | 4   | 53.07   | 14.18  | 32.75  |
> | 4096  | 14336 | 2   | 34.70   | 15.74  | 49.56  |
> | 4096  | 14336 | 3   | 38.09   | 11.80  | 50.11  |
> | 4096  | 14336 | 4   | 49.33   | 13.61  | 37.07  |
>
> > Q2. Have the authors tested larger models, like Qwen3-32B or Llama3-70B? It is often observed that some optimizations may perform differently in these larger models.
> - Thank you for this helpful suggestion. We fully agree that evaluating larger models is important for validating the generality of our approach. Following your comment, we additionally evaluated Qwen3-32B with AnyBCQ and observed that our method remains applicable and effective even at this larger scale.
> - As shown below, AnyBCQ at 3 and 4 bits achieves accuracy close to FP16 on WikiText, C4, and MMLU, while 2 bit AnyBCQ still provides a reasonable trade off between compression and accuracy. This trend is consistent with our results on smaller models and supports the claim that AnyBCQ scales well to larger LLMs.
>
> | Model     | Method | Bit | Wiki PPL | C4 PPL | MMLU (5-shot) |
> | --------- | ------ | --- | -------- | ------ | ------------- |
> | Qwen/Qwen3-32B | FP16   | 16  | 7.61     | 12.72  | 0.8185        |
> | Qwen/Qwen3-32B | AnyBCQ | 2   | 12.07    | 16.96  | 0.7070        |
> | Qwen/Qwen3-32B | AnyBCQ | 3   | 8.42     | 13.62  | 0.7972        |
> | Qwen/Qwen3-32B | AnyBCQ | 4   | 7.81     | 13.06  | 0.8070        |

---

> > ### Author Response · Authors · 2025-11-21
> >
> > > Q3. The authors claim that the performance gap increases as matrix sizes grow, but the throughput gains seem fairly constant on end-to-end evaluations (+/- 20 tokens per second). How is this rationalized?
> > - Thank you for raising this point. To clarify the trend, we additionally measured end-to-end throughput (tokens/sec) for both Llama-3.1-8B and Llama-3.1-70B. These results show that, while the absolute throughput difference in tokens/sec remains on the order of 10–20 tokens/sec, the relative improvement indeed grows with model size: approximately 1.07–1.10× for the 8B model versus 1.23–1.29× for the 70B model. This aligns with our claim that larger matrices benefit more from our method, since GEMM dominates a larger fraction of the end-to-end latency as the model scales up. We have clarified this distinction between kernel-level scaling behavior and end-to-end absolute token/s differences in the revised manuscript.
> >
> > | Model         | Bit | AnyBCQ | Any-Precision LLM | Improvement |
> > | ------------- | --- | ------ | ----------------- | ----------- |
> > | Llama-3.1-8B  | 2   | 245    | 228  | 1.07× |
> > | Llama-3.1-8B  | 3   | 212    | 196  | 1.08× |
> > | Llama-3.1-8B  | 4   | 186    | 169  | 1.10× |
> > | Llama-3.1-70B | 2   | 50     | 39   | 1.29× |
> > | Llama-3.1-70B | 3   | 40     | 32   | 1.23× |
> > | Llama-3.1-70B | 4   | 33     | 26   | 1.25× |

---

> > > ### Comment · Reviewer_fpbz · 2025-11-24
> > >
> > > Thank you for the response. As your clarification and additional experiments have resolved part of my concerns, I am still inclined to recommend acceptance and will keep my score.

---

> > > > ### Author Response · Authors · 2025-11-25
> > > >
> > > > Thank you very much for your thoughtful follow-up and for your positive recommendation. Your comments and suggestions were very helpful in improving the clarity and quality of our paper, and we truly appreciate the time and care you put into your review. If you have any further questions or suggestions, please feel free to let us know.

---

### Official Review · Reviewer_d4yW · 2025-10-31

**Soundness:** 3
**Presentation:** 2
**Contribution:** 3
**Rating:** 4
**Confidence:** 4

**Summary:**

This paper introduces AnyBCQ, a multi-precision quantization framework for LLMs based on Binary-Coded Quantization (BCQ). The core idea is to share binary bit-planes across precisions while storing precision-specific scale factors, enabling hardware-efficient, dynamic-precision inference.

**Strengths:**

1. The central idea of extending BCQ to a multi-precision setting via shared binary codes and specialized, per-precision scales is novel and interesting.
2. Figures (e.g., Figure 1, Figure 3) are effective in visualizing the method's mechanics and differentiating it from prior non-uniform quantization work.
3. The co-design of the quantization algorithm with a hardware-efficient CUDA kernel, which avoids the typical overheads of non-uniform methods, is a strength.

**Weaknesses:**

1. The paper's core contribution, sharing binary codes while maintaining separate scales, is not highlighted sufficiently early. The introduction should be more compact and should proactively introduce the fundamentals of BCQ. Many readers may be more familiar with index-based quantization (e.g., AWQ/GPTQ), and this method is different. Visualizing the storage savings (from Table 1) earlier in the paper could help ground this contribution.

2. In Table 2, the "Proposed (fixed-precision)" baseline outperforms ShiftAddLLM, especially at 2/3 bits. The paper needs to clarify if "fixed-precision" is a plain BCQ implementation or if it includes additional optimizations not present in ShiftAddLLM.

3. ShiftAddLLM is absent from the end-to-end performance evaluations in Table 4 and Figure 4. Please provide.

4. The "fixed-precision" model, lacking the shared-binary constraint, should theoretically always outperform the "multi-precision" model. However, in Table 2, the multi-precision model occasionally scores higher. Please explain. Providing perplexity scores (C4/Wikitext) for both fixed- and multi-precision models would also be beneficial.

5. The motivation for multi-precision models is dynamic, runtime adaptation (e.g., token-level precision switching). However, all evaluations are static (i.e., the entire inference is run at a single, fixed bit-width). The paper would be much stronger if it included a case study demonstrating a practical scenario that achieves a better accuracy/latency trade-off than any single static precision.

6. The paper's most significant gains are in the 2-bit regime. My major concern is about practical utility. As shown in Table 4, the 2-bit perplexity is extremely high (e.g., 19.01 for Llama-8B), rendering the model practically unusable for coherent generation. Given that 4-bit quantization is acknowledged as nearly lossless with tolerable latency, the paper should provide a compelling use case or scenario where such a high-perplexity 2-bit model would actually be deployed. Or, illustrate scenarios where parts of the response could be generated at 2-bit precision without significantly harming overall quality.

**Questions:**

Please see the weaknesses section and clarify these points in your rebuttal.

---

> ### Author Response · Authors · 2025-11-20
>
> > W1. The paper's core contribution, sharing binary codes while maintaining separate scales, is not highlighted sufficiently early. The introduction should be more compact and should proactively introduce the fundamentals of BCQ. Many readers may be more familiar with index-based quantization (e.g., AWQ/GPTQ), and this method is different. Visualizing the storage savings (from Table 1) earlier in the paper could help ground this contribution.
> - Thank you for the helpful suggestion. We agree that sharing binary codes while maintaining separate scales is the central idea of our method, and we now highlight this more clearly in the revised manuscript. Specifically, we have (i) moved the storage comparison (originally Table 1) into the introduction and refer to it when introducing our method, and (ii) added a brief primer on BCQ in the introduction, explicitly contrasting it with index-based quantization methods (e.g., AWQ/GPTQ) to clarify how our approach differs. We appreciate your constructive feedback.
>
> > W2. In Table 2, the "Proposed (fixed-precision)" baseline outperforms ShiftAddLLM, especially at 2/3 bits. The paper needs to clarify if "fixed-precision" is a plain BCQ implementation or if it includes additional optimizations not present in ShiftAddLLM.
> - Thank you for pointing this out, and we apologize for the confusion. Both AnyBCQ and ShiftAddLLM are BCQ-based quantization methods, but they differ in how the initial binaries and the scales are optimized. ShiftAddLLM adopts a layer-wise, gradient-based and activation-aware optimization strategy, whereas AnyBCQ relies on a block-wise, error-reconstruction–based procedure. We have revised the manuscript to clarify this distinction and to explicitly describe what is included in the "fixed-precision" baseline.

---

> > ### Author Response · Authors · 2025-11-20
> >
> > > W3. ShiftAddLLM is absent from the end-to-end performance evaluations in Table 4 and Figure 4. Please provide.
> > - Thank you for the suggestion. We have added ShiftAddLLM (SAL) to our end to end evaluation and report Wiki perplexity, MMLU accuracy, and throughput. Since SAL is a BCQ based fixed precision method, it can reuse the same CUDA kernel as AnyBCQ, so SAL and AnyBCQ achieve essentially identical throughput across all bit settings while differing only in the quantization procedure. Empirically, SAL shows comparable or slightly better accuracy than AnyBCQ in the 4 bit regime, while AnyBCQ provides the additional benefit of supporting multi precision operation from the same set of shared binary codes. The updated figures in the revised manuscript make this trade off explicit by jointly comparing FP16, AnyPrecision LLM, ShiftAddLLM, and AnyBCQ.
> >
> > | Model        | bit | Wiki FP16 | Wiki AP  | Wiki SAL | Wiki AB |
> > |--------------|-----|-----------|----------|----------|---------|
> > | Llama-3.1-8B | 2   | 6.24      | 1680.77  | 69.87    | 19.01   |
> > | Llama-3.1-8B | 3   | 6.24      | 8.60     | 8.88     | 8.08    |
> > | Llama-3.1-8B | 4   | 6.24      | 6.70     | 6.76     | 6.84    |
> > | Gemma-2-9b   | 2   | 6.84      | 19.62    | 27.38    | 12.72   |
> > | Gemma-2-9b   | 3   | 6.84      | 7.94     | 8.68     | 7.95    |
> > | Gemma-2-9b   | 4   | 6.84      | 7.06     | 7.22     | 7.23    |
> > | Phi-4-14b    | 2   | 6.46      | 13.37    | 14.69    | 9.97    |
> > | Phi-4-14b    | 3   | 6.46      | 7.14     | 7.21     | 7.44    |
> > | Phi-4-14b    | 4   | 6.46      | 6.61     | 6.59     | 7.16    |
> >
> > | Model        | bit | MMLU FP16 | MMLU AP | MMLU SAL | MMLU AB |
> > |--------------|-----|-----------|---------|----------|---------|
> > | Llama-3.1-8B | 2   | 0.6535    | 0.2466  | 0.2670   | 0.3532  |
> > | Llama-3.1-8B | 3   | 0.6535    | 0.5553  | 0.5785   | 0.5828  |
> > | Llama-3.1-8B | 4   | 0.6535    | 0.6404  | 0.6357   | 0.6315  |
> > | Gemma-2-9b   | 2   | 0.7074    | 0.3309  | 0.2678   | 0.4336  |
> > | Gemma-2-9b   | 3   | 0.7074    | 0.6530  | 0.6114   | 0.6538  |
> > | Gemma-2-9b   | 4   | 0.7074    | 0.6923  | 0.6847   | 0.6881  |
> > | Phi-4-14b    | 2   | 0.8039    | 0.4952  | 0.5635   | 0.6638  |
> > | Phi-4-14b    | 3   | 0.8039    | 0.7732  | 0.7756   | 0.7708  |
> > | Phi-4-14b    | 4   | 0.8039    | 0.7975  | 0.7973   | 0.7904  |
> >
> > | Model        | bit | Wiki FP16 | Token/sec FP16 | Token/sec AP | Token/sec SAL | Token/sec AB |
> > |--------------|-----|-----------|----------------|-------------|--------------|--------------|
> > | Llama-3.1-8B | 2   | 6.24      | 105            | 228         | 245          | 245          |
> > | Llama-3.1-8B | 3   | 6.24      | 105            | 196         | 212          | 212          |
> > | Llama-3.1-8B | 4   | 6.24      | 105            | 169         | 186          | 186          |
> > | Gemma-2-9b   | 2   | 6.84      | 83             | 163         | 185          | 185          |
> > | Gemma-2-9b   | 3   | 6.84      | 83             | 144         | 164          | 164          |
> > | Gemma-2-9b   | 4   | 6.84      | 83             | 125         | 146          | 146          |
> > | Phi-4-14b    | 2   | 6.46      | 56             | 147         | 171          | 171          |
> > | Phi-4-14b    | 3   | 6.46      | 56             | 125         | 144          | 144          |
> > | Phi-4-14b    | 4   | 6.46      | 56             | 105         | 123          | 123          |

---

> > > ### Author Response · Authors · 2025-11-20
> > >
> > > > W4. The "fixed-precision" model, lacking the shared-binary constraint, should theoretically always outperform the "multi-precision" model. However, in Table 2, the multi-precision model occasionally scores higher. Please explain. Providing perplexity scores (C4/Wikitext) for both fixed- and multi-precision models would also be beneficial.
> > > - Thank you for pointing this out, and we apologize for the confusion. As you correctly noted, a fixed-precision model without the shared-binary constraint should theoretically dominate the multi-precision model. In Table 2, the cases where the multi-precision model slightly outperforms the fixed-precision one occur only when the performance gap is very small, so a few sub-tasks can flip due to evaluation noise and task-level variance rather than a systematic advantage.
> > > Following your suggestion, we have added perplexity results on WikiText and C4 for both fixed- and multi-precision models in the revised manuscript:
> > >
> > > | Method   | Bit | Wiki ppl | C4 ppl |
> > > |---|-----|-----|---|
> > > | Proposed (Fixed-prec.) | 2 | 18.95 | 21.64 |
> > > | Proposed (Multi-prec.) | 2 | 19.01 | 21.58 |
> > > | Proposed (Fixed-prec.) | 3 | 7.68  | 12.23 |
> > > | Proposed (Multi-prec.) | 3 | 8.08  | 12.41 |
> > > | Proposed (Fixed-prec.) | 4 | 6.62  | 10.26 |
> > > | Proposed (Multi-prec.) | 4 | 6.84  | 10.65 |
> > > - These results are consistent with the theoretical expectation: since the 2-bit model is used as the base, the fixed- and multi-precision models behave almost identically at 2 bits, while at higher bit-widths the additional shared-binary constraint makes the multi-precision model slightly worse than the fixed-precision model. We have clarified this discussion in the revised version.
> > >
> > > > W5. The motivation for multi-precision models is dynamic, runtime adaptation (e.g., token-level precision switching). However, all evaluations are static (i.e., the entire inference is run at a single, fixed bit-width). The paper would be much stronger if it included a case study demonstrating a practical scenario that achieves a better accuracy/latency trade-off than any single static precision.
> > >
> > > - Thank you for the helpful suggestion. We agree that a case study demonstrating a practical scenario would clarify our motivation and strengthen the paper. Following your suggestion, we have added PMPD results using our anybcq model. PMPD [1] is a method that performs mixed-precision inference with multi-precision models. We evaluate it on the CNN/DailyMail (CNN/DM) summarization task.
> > > - The results align with our expectations. Since AnyBCQ outperforms AnyPrecision, especially in lower-bit configurations, PMPD with AnyBCQ achieves higher CNN/DM ROUGE-L and BERTScore than PMPD with AnyPrecision under the same average bit precision.
> > > | Method       | Average Precision | ROUGE-L  | BERTScore |
> > > |-------|----|---:|-----------:|
> > > | Any-Precision LLM | 3.6 | 0.154   | 0.840   |
> > > | AnyBCQ      | 3.6  | 0.178 | 0.849  |
> > > | Any-Precision LLM | 3.15      | 0.148   | 0.836  |
> > > | AnyBCQ      | 3.15   | 0.155 | 0.843 |
> > > | Any-Precision LLM | 2.23       | 0.097   | 0.821  |
> > > | AnyBCQ      | 2.23    | 0.113 | 0.830  |
> > >
> > >
> > > [1] Chen, Hao Mark, et al. "Progressive Mixed-Precision Decoding for Efficient LLM Inference." The Thirteenth International Conference on Learning Representations (ICLR). 2025.
> > >
> > >
> > > > W6. The paper's most significant gains are in the 2-bit regime. My major concern is about practical utility. As shown in Table 4, the 2-bit perplexity is extremely high (e.g., 19.01 for Llama-8B), rendering the model practically unusable for coherent generation. Given that 4-bit quantization is acknowledged as nearly lossless with tolerable latency, the paper should provide a compelling use case or scenario where such a high-perplexity 2-bit model would actually be deployed. Or, illustrate scenarios where parts of the response could be generated at 2-bit precision without significantly harming overall quality.
> > >
> > > - Thank you for the helpful suggestion. We agree with the reviewer that a purely 2-bit model has limited practical utility for open-ended generation. As noted, 4-bit quantization is almost lossless in many settings, whereas 2-bit quantization incurs substantially larger errors. In particular, for the 2-bit case, our intention is not to deploy a stand-alone 2-bit model, but rather to use 2-bit precision within mixed-precision inference scenarios. In these settings, recent studies such as PMPD demonstrate the effectiveness of mixed-precision inference by gradually lowering the precision along the generated sequence. These works show that using 3-bit or 2-bit precision for selected parts of the model or computation can yield additional efficiency gains over uniform 4-bit quantization, while keeping the degradation in quality small. Our contribution is to develop a multi-precision model that is well suited for these mixed-precision deployments, achieving benefits beyond 4-bit-only inference while minimizing the performance loss from the 2-/3-bit components.

---

> > > > ### Comment · Reviewer_d4yW · 2025-11-24
> > > >
> > > > thanks for addressing my concern in a very detailed manner. no more questions at this time.

---

> > > > > ### Author Response · Authors · 2025-11-25
> > > > >
> > > > > Thank you very much for your thoughtful follow-up and for your positive recommendation. Your comments and suggestions were very helpful in improving the clarity and quality of our paper, and we truly appreciate the time and care you put into your review. If you have any further questions or suggestions, please feel free to let us know.

---

### Author Response · Authors · 2025-11-20

We sincerely thank the reviewers for their valuable feedback. In the revised manuscript, we have adopted the following color scheme to make it easier to identify changes made in response to each reviewer:

Reviewer d4yW – Orange,

Reviewer fpbz – Blue,

Reviewer iC2H – Red,

Reviewer 7UfK – Violet.

---

### Author Response · Authors · 2025-12-03

We apologize for the situation arising from the recent incident and recognize that it may impose an additional strain on the incoming Area Chair. We sincerely appreciate your willingness to assume this role at a late stage and to review our submission despite these exceptional circumstances.
To facilitate your evaluation and to clarify how our response addresses the major critiques, we provide the table below summarizing the key review points alongside our rebuttals.

### Reviewer: d4yW (Score 4 -> 6)
- | Weakness & Question | Author Rebuttal |
|---|---|
| **Core contribution not highlighted early; BCQ basics unclear vs AWQ/GPTQ.** | Moved the storage comparison into the introduction and added a short BCQ primer + explicit contrast to index-based quantization (AWQ/GPTQ). |
| Clarify why “Proposed (fixed-precision)” beats ShiftAddLLM at 2/3 bits | Clarified both are BCQ-based but differ in optimization: ShiftAddLLM uses layer-wise, gradient/activation-aware optimization; AnyBCQ uses block-wise error-reconstruction. Updated manuscript to explain the fixed-precision baseline precisely. |
| **ShiftAddLLM missing from end-to-end results.** | Added ShiftAddLLM to end-to-end eval (Wiki perplexity, MMLU, token/sec). Noted throughput is essentially identical (same kernel), with differences mainly from quantization procedure; emphasized AnyBCQ’s advantage is multi-precision from shared bit-planes. |
| Why can multi-precision sometimes slightly outperform fixed-precision (shouldn’t it be dominated)? | Attributed to evaluation noise/task variance when gaps are tiny; added WikiText + C4 perplexity for fixed vs multi precision showing expected trend (multi-precision slightly worse at higher bits due to shared-binary constraint). |
| **Multi-precision motivation is dynamic switching, but evaluations were static; need a real case study.** | Added a mixed-precision decoding case study (PMPD) on CNN/DailyMail; showed AnyBCQ improves ROUGE-L and BERTScore vs Any-Precision LLM under the same average precision. |
| Practical utility concern: 2-bit perplexity is very high (e.g., Llama-8B), seems unusable. | Argued 2-bit is not intended as a standalone setting; the target is mixed-precision inference where some parts can be 2/3-bit for extra efficiency while maintaining quality (e.g., PMPD-style schedules). |

### Reviewer: fpbz (Score 6 -> 6)
- | Weakness & Question | Author Rebuttal |
|---|---|
| Kernel analysis too high-level: **latency breakdown (bit-transpose vs LUT lookup)** needed. | Instrumented Any-Precision LLM kernel using `clock64()` and reported component breakdown: bit-transpose is dominant (~35–58%), LUT lookup ~9–17%, remainder GEMM/memory; added to revised manuscript. |
| Test larger models. | Added Qwen3-32B results (Wiki/C4 perplexity, MMLU) showing AnyBCQ remains effective. |
| Throughput gains look roughly constant in tokens/sec; why claim gap increases with matrix size? | Clarified that while absolute token/sec deltas can be similar, **relative** speedup grows for larger models (e.g., ~1.07–1.10× at 8B vs ~1.23–1.29× at 70B) because GEMM dominates more at scale. |

### Reviewer: iC2H (Score 8 -> no answer)
- | Weakness & Question | Author Rebuttal |
|---|---|
| Experiments only on A100; validate on other hardware. | Added H100 end-to-end throughput; reported speedups are preserved across A100/H100 and discussed relevance to variable-bit accelerators (BCQ-friendly designs). |
| Missing energy / memory-bandwidth evidence. | Added nvidia-smi–based measurements (power, DRAM traffic proxies like mem util, latency) on a GEMM workload; reported lower power and latency for AnyBCQ vs Any-Precision LLM at 2/3/4 bits and higher memory utilization. |
| Lacks theoretical depth/formal conditions. | Acknowledged as a limitation; added a discussion framing theoretical analysis as future work (esp. progressive precision expansion dynamics). |

### Reviewer: 7UfK (Score 6 -> 8)
- | Weakness & Question | Author Rebuttal |
|---|---|
| Why BCQ can be better/worthwhile vs non-uniform quantization (e.g., SqueezeLLM)? Is it SOTA-competitive? | Explained representational tradeoff: BCQ is more expressive than uniform (per-plane scales) but more hardware-regular than fully non-uniform; effectiveness comes from block-wise MSE error reconstruction. Noted SqueezeLLM’s hybrid FP16/sparse design is less kernel-friendly; cited BCQ-based ShiftAddLLM results as competitive with SOTA low-bit methods. |
| Generative task evaluation (e.g., HumanEval) missing. | Added a generative-style evaluation via CNN/DailyMail summarization using PMPD; reported ROUGE-L and BERTScore improvements for AnyBCQ under matched average precision. |
| Outlier where AnyBCQ is slower than AnyPrecision LLM for one shape / modest speedup explanation. | Attributed outlier to non-exhaustive kernel retuning (tile size choice can be suboptimal for some shapes); reiterated modest GPU gains but emphasized larger expected benefits on native variable-bit accelerators. |

---

> ### Author Response · Authors · 2025-12-03
>
> During the rebuttal period, we conducted additional experiments and qualitative analyses that, in our view, more clearly articulate and further substantiate the contributions of AnyBCQ. Reflecting these clarifications and new evidence, reviewers d4yW and 7UfK increased their overall assessments from 4 to 6 and from 6 to 8, respectively, after considering our supplementary results and responses.
>
> **To the best of our knowledge, AnyBCQ is the first approach to generalize Binary-Coded Quantization (BCQ) to a multi-precision setting, thereby enabling hardware-friendly mixed-precision inference.** We believe this any-precision LLM model provides a practical basis for deploying multi-precision LLMs under a range of service-level objectives and operational constraints.
>
> Finally, we would like to reiterate our gratitude to the Area Chair for the time and careful attention devoted to evaluating our work under challenging circumstances. We also appreciate the committee’s consideration of AnyBCQ, and we hope this summary is useful in informing the final decision.

---

### Meta-Review · Area_Chair_8aTy · 2026-01-07

**Summary:**

This paper proposes a sensible and effective approach for quantization of LLMs bit planes. The idea basically extends BCQ to the mixed-precision case, which can enable more fine-grained quantization and better explore the efficiency/performance frontier. The approach is tested on

Reviewers generally noted that the idea is good, and results in practical improvements. Reviewers were particularly appreciative of the hardware-efficient kernel associated with the the algorithm. On the negative side, there were some concerns with regard to scope: in particular the model was only tested on an A100, and with a small model. Moreover, some of the initial analyses could have been more comprehensive. However, these weaknesses were adequately addressed during the rebuttal.

**Reviewer Concerns:**

Reviewer concerns included:
- Experiments conducted only on specific hardware (A100)
- Experiments on a smaller model
- Analysis/numbers at a high level (e.g., no memory bandwidth numbers)

These were all adequately addressed during the rebuttal.

**Reviewer Scores:**

Based on the rebuttal:
Reviewer d4yW chose to increase score to 6
Reviewer fpbz chose to maintain their score
Reviewer 7UfK increased their score to 8

Reviewer iC2H did not answer, but my sense is that they would have kept their score of 8.

---

### Decision · Program_Chairs · 2026-01-26

Accept (Poster)